# Endogenous Glycoprotein GPM6a Is Involved in Neurite Outgrowth in Rat Dorsal Root Ganglion Neurons

**DOI:** 10.3390/biom13040594

**Published:** 2023-03-25

**Authors:** Gabriela I. Aparicio, Antonella León, Rocío Gutiérrez Fuster, Baylen Ravenscraft, Paula V. Monje, Camila Scorticati

**Affiliations:** 1Department of Neurosurgery, College of Medicine, University of Kentucky, Lexington, KY 40536-0298, USA; 2Brain Restoration Center, College of Medicine, University of Kentucky, Lexington, KY 40536-0298, USA; 3Instituto de Investigaciones Biotecnológicas, Universidad Nacional de San Martín (UNSAM), Consejo Nacional de Investigaciones Científicas y Técnicas (CONICET), San Martín, Buenos Aires 1650, Argentina; 4Escuela de Bio y Nanotecnologías (EByN), Universidad Nacional de San Martín, San Martín, Buenos Aires 1650, Argentina; 5Department of Neurological Surgery, Stark Neuroscience Research Institute, Indiana University School of Medicine, Indianapolis, IN 46202, USA

**Keywords:** GPM6A, glycoprotein, dorsal root ganglion neurons, peripheral nervous system, neuritogenesis

## Abstract

The peripheral nervous system (PNS) has a unique ability for self-repair. Dorsal root ganglion (DRG) neurons regulate the expression of different molecules, such as neurotrophins and their receptors, to promote axon regeneration after injury. However, the molecular players driving axonal regrowth need to be better defined. The membrane glycoprotein GPM6a has been described to contribute to neuronal development and structural plasticity in central-nervous-system neurons. Recent evidence indicates that GPM6a interacts with molecules from the PNS, although its role in DRG neurons remains unknown. Here, we characterized the expression of GPM6a in embryonic and adult DRGs by combining analysis of public RNA-seq datasets with immunochemical approaches utilizing cultures of rat DRG explants and dissociated neuronal cells. M6a was detected on the cell surfaces of DRG neurons throughout development. Moreover, GPM6a was required for DRG neurite elongation in vitro. In summary, we provide evidence on GPM6a being present in DRG neurons for the first time. Data from our functional experiments support the idea that GPM6a could contribute to axon regeneration in the PNS.

## 1. Introduction

The peripheral nervous system (PNS) has the unique capacity for functional recovery and self-repair after an injury, a feature seemingly missing from the central nervous system (CNS) [1]. Both the intrinsic capacity of PNS neurons to re-initiate axon growth and the reprogramming features of PNS-resident glial cells contribute to nerve regeneration after injury [2,3]. Indeed, the regenerative properties of PNS cells have been exploited to develop therapies to treat CNS trauma after spinal cord injury and prevent neurodegeneration in Parkinson’s disease [4,5,6,7,8,9,10]. However, the complete molecular mechanisms underlying PNS regeneration have not been elucidated yet.

Peripheral sensory neurons residing in the dorsal root ganglia (DRG), interconnect the central and the peripheral nervous systems. DRG neurons are pseudo-unipolar; they possess a single axon extending from the cell body, and then it bifurcates into a central axonal branch (to the spinal cord) and a peripheral axonal branch to their target tissues, [11,12]. After an injury, DRG neurons induce axon regeneration by adjusting the expression of cell adhesion molecules, neurotrophins (NTs), growth factors receptors, and cytoskeletal molecules, among others [13]. For instance, NTs, nerve growth factor (NGF), brain-derived neurotrophic factor (BDNF), and neurotrophin-3 (NT3), along with their cognate receptors (TrKA, TrKB, and TrKC), are critical for the survival, development, and maintenance of DRG neurons [14,15]. However, the complete molecular environment that allows the DRG development program to switch on still needs to be unveiled.

The neuronal membrane glycoprotein GPM6a has emerged as a novel molecule involved in neuronal development and structural plasticity in the CNS, and was proposed as a target gene in various neuropsychiatric disorders [16]. GPM6a belongs to the proteolipid protein (PLP) family together with PLP1, DM20, and GPM6b [17]. PLP family members share structural similarity with the tetraspanin protein family [18] and have four transmembrane domains, small (EC1) and large (EC2) extracellular domains, one intracellular domain, and the N- and C-terminus within the cell cytoplasm. GPM6a is highly expressed in the hippocampus, cerebellum, striatum, and prefrontal cortex, among other brain areas. Specifically, GPM6a is mostly located on the cell surfaces of the neurons and epithelial cells of the choroid plexus, but not on glial cells. In contrast, PLP1 and DM20 are expressed in glial cells only, but GPM6b is expressed in both neurons and glial cells [19]. In vitro studies demonstrated that GPM6a promotes neurite extension, filopodia/spine induction, and synapse formation and maintenance [20,21,22,23,24,25]. Amino acids within M6a´s primary sequence that are critical for its function have been described (reviewed in [16]); in particular, both of GPM6a’s extracellular domains are critical for GPM6a protein-protein interactions and thus are functional domains. Indeed, blocking GPM6a extracellular domains with a monoclonal structural antibody (GPM6a-mAb) arrests neurite extension and synapse formation in neurons from embryonic hippocampi [18,26] and cerebellar explants [27].

We hypothesized that GPM6a interacts with different proteins through its extracellular domains, thereby triggering different responses within neurons. In this regard, we recently identified 72 proteins that could potentially interact with the extracellular domains of GPM6a. Roughly 60% of these proteins are located on the membranes of the synaptic compartments, and a few of them were validated (Syn1, SYP, NMDA-R1, SV2B). Interestingly, ~30% of the potentially interacting proteins are glial-specific not only for oligodendrocytes but also for Schwann cells (SCs), such as PLP1, which has already been validated [16,28]. Indeed, Bang and colleagues identified GPM6b as a component of Schwann cell microvilli that stabilizes the nodes of Ranvier by its association with glial gliomedin, glial NrCAM, and axonal NF186 [29]. Dataset libraries such as BioGPS http://biogps.org; accessed on 21 June 2021 [30] and the Human Protein Atlas https://www.proteinatlas.org; accessed on 21 June 2021 [31] show the expression of the human GPM6A gene in DRG neurons. Altogether, GPM6a could be expressed in DRG neurons, interacting with neuronal proteins, and also with glial proteins located on the cell membranes of SCs to promote neuron development and repair. Therefore, the purpose of this study was to investigate the expression and function of GPM6a in DRG neurons and explants from both embryos and adult rats.

## 2. Materials and Methods

### 2.1. Animals

Six-week-old and pregnant Sprague–Dawley rats were purchased from Facultad de Veterinaria of the Universidad of Buenos Aires (FVet-UBA) or Charles River (Chicago, IL, USA), and they were maintained on a 12/12-h light/dark cycle with food and water available ad libitum. All animal procedures were carried out according to the guidelines of the National Institutes of Health (publications 80–23), the Guide for the Care and Use of Laboratory Animals (National Research Council), and the Guidelines of the Institutional Animal Care and Use Committee of the Indiana University School of Medicine.

### 2.2. Reagents and Antibodies

Monoclonal primary antibodies used were anti-GPM6a rat IgG (GPM6a-mAb, 1/1000, Medical and Biological Laboratories, MBL, Nagoya, Japan), anti-β-tubulin III mouse IgG (Tuj-1-mAb, 1/1000, Covance, Princeton, NJ, USA), and anti-alpha-tubulin mouse IgG (anti-Tub, Sigma, Munich, Germany). Monoclonal antibodies produced in-house from hybridoma cell cultures were anti-neurofilament heavy chain mouse IgG (NFh-mAb), hybridoma cell line RT97 (DSHB, Iowa City, IA, USA), and anti-Thy-1 mouse IgM (Thy-1-mAb) (ATCC, TIB-103). Polyclonal primary antibodies were rabbit anti-C terminus of GPM6a (anti-GPM6a) developed in our laboratory [23] and polyclonal anti-S100β rabbit IgG (1/500, Dako, Carpinteria, CA, USA). F-Actin filaments were stained with rhodamine or fluorescein isothiocyanate-conjugated phalloidin (1/1000, Molecular Probes, Eugene, OR, USA). Pre-absorbed secondary antibodies were Alexa Fluor-488 goat anti-rabbit IgG, goat anti-rat IgG, and goat anti-mouse IgG; Alexa Fluor-568 goat anti-mouse IgG; and Alexa Fluor-594 goat anti-rat IgG and goat anti-mouse IgM (Invitrogen, Eugene, OR, USA). All secondary antibodies were used at 1/1000. Polyclonal HRP-conjugated antibodies were goat anti-rabbit IgG and goat anti-mouse IgG (Sigma, St. Louis, USA). Other reagents used were methyl green (Sigma), paraformaldehyde 16% solution (PFA, Electron Microscopy Sciences, PA, USA), Mowiol^®^ 4-88 (Sigma), and protease inhibitor cocktail (PIC, Sigma). Unless specified within the text, all antibodies and reagents were prepared in PBS.

### 2.3. Embryonic DRG Explants and Dissociated DRG Neuronal Cultures

Cultures of embryonic DRG explants and dissociated DRG neurons were established from rat embryos between 15 and 18 days of gestation used for Figure 1a,b and Figure 2a–c. The DRG bodies were extracted from the spinal cord with fine forceps and then treated according to the experiment to be conducted. For DRG explant cultures, embryonic DRG bodies were placed into the center of the well of a 24-well plate coated with 0.1 mg/mL poly-L-lysine hydrobromide (PLL, Sigma) and 20 mg/mL Laminin (Invitrogen) and left to sit on the bottom of the well before adding the culture media. In the case of dissociated DRG neuron cultures, the DRG bodies were dissociated with 0.25% trypsin for 45 min and placed in a rotating incubator (not gassed) at 37 °C, followed by gentle trituration. The resulting cell suspensions were seeded on coverslips coated with PLL-Laminin at a low density, 3 × 10^4^ cells/well, into a 24-well plate. Both types of cultures were established and maintained at 37 °C and 5% CO_2_ in Neurobasal^TM^ medium containing B27 supplement (Thermo Fisher Scientific, Waltham, MA, USA), 25 ng/mL nerve growth factor (R&D Systems, Minneapolis, MN), 1 mM L-glutamine (Thermo Fisher Scientific, Waltham, MA, USA), 1% glucose, and 10% fetal horse serum (Sigma).

### 2.4. Isolation and Culture of Dissociated DRG Neurons from Adult Rats

DRGs and dissociated DRG neurons from 6–8-week-old Sprague–Dawley rats were used for Figure 1c,d, Figure 3f, and Appendix A. Both male and female animals were used in approximately equal numbers for all of the experiments. Each experiment contained a minimum of three independent experiments. Adult DRG neurons cultures were prepared according to the method described by Burkey et al. [32]. Briefly, rats were sacrificed and the DRGs were dissected and collected in ice-cold modified Hank’s balance salt solution (HBSS, 171 mM NaCl, 6.7 mM KCL, 1.6 mM Na_2_PO_4_, 0.5 mM KH_2_PO_4_, 6 mM D-glucose, pH 7.3) supplemented with 50 U/mL penicillin/streptomycin (Sigma). HBSS was carefully aspirated and replaced with a pre-warmed collagenase type 1A solution (1 mL per 6 isolated DRGs) and incubated 30 min at 37 °C. Then, an additional digestion for 30 min in 1 mL of 0.25% trypsin in HBSS was carried out at 37 °C according to Kim et al. [33]. The supernatant was aspirated, and the DRG pellet was resuspended in adult growth medium (AGM, Ham’s F12 medium (GIBCO, Thermo Fisher Scientific, Waltham, MA, USA) containing 10% horse serum, 2 mM L-glutamine, and 50 U/mL penicillin/streptomycin. The DRG cells were dissociated by gentle mechanical agitation using a glass Pasteur pipet until the suspension became homogenous. Cells were collected by centrifugation at 200× *g* for 10 min, and AGM with NGF (250 ng/mL) was added for cell counting and plating. Cells were seeded directly onto coverslips coated with PLL-Laminin at a low density (approx. 10–20 × 10^4^ cells/well) in a 24-well plate. Cells were maintained at 37 °C and 5% CO_2_ with medium replacement every 2 days.

For the DRG neurite outgrowth assay, embryonic and adult dissociated DRG neurons were cultured in 24-well plates at low density, (10 × 10^4^ cells/well), for 24 h in the presence of GPM6a-mAb (1 µg/mL and 3 µg/mL) or PBS (control condition). For neurite outgrowth measurements, N = 2–4 independent experiments were performed, and each experiment had at least three wells per condition (n = 3 technical replicates).

### 2.5. Immunohistochemistry of DRG Explants

Embryonic and adult DRGs were fixed in 4% PFA-PBS for 24 h, cryoprotected with 30% sucrose (Sigma) for 48 h, replaced by Cryoplast^®^ (Biopack, Zarate, Bs. As., Argentina), and frozen in dry ice. To perform immunohistochemistry, 20 µm of DRG sections (Cryostat Leica CM1860, Leica Biosystems, Nußloch, Alemania) were adhered to positively charged slides (Biotraza, Bs. As., Argentina) and maintained at −80 °C until use. Sections were dried at RT for 20 min, rinsed with PBS, permeabilized with 0.25% Triton X-100 for 5 min, and blocked with 10% horse serum for 1 h. Then, sections were incubated with GPM6a-mAb and Tuj1-mAb antibodies in 3% bovine serum albumin (BSA) for 48 h at 4 °C. Subsequently, the sections were rinsed with PBS and labeled with the corresponding pre-absorbed secondary antibodies for 1 h at RT. Cell nuclei were stained with methyl green (1/10,000) and mounted with coverslips in Mowiol^®^.

Embryonic DRG explants were fixed in 4% PFA-PBS at 10 or 21 DIV (days in vitro) (Figure 2) before permeabilizing and blocking, as described above. Subsequently, DRG explants were incubated with GPM6a-mAb together with NFh-mAb, Thy-1-mAb, or S100β antibodies in 3% BSA for 16 h at 4 °C. Lastly, the explants were rinsed with PBS, labeled with the corresponding pre-absorbed secondary antibodies for 1 h at RT, and mounted with coverslips in Mowiol^®^ right after staining the cell nuclei with DAPI.

### 2.6. Immunofluorescence in Dissociated DRG Neurons

For endogenous GPM6a staining, dissociated DRG neurons were incubated for 1 h at 4 °C with GPM6a-mAb in fresh medium. Afterward, cells were washed with PBS and labeled with Alexa Fluor-488 goat anti-rat IgG for 1 h at 4 °C. Then, the cells were fixed in 4% PFA and 4% sucrose at RT for 10 min, permeabilized with 0.1% Triton X-100 for 5 min, blocked with 3% BSA for 1 h, and labeled with Tuj-1-mAb antibodies in 1% BSA for 16 h at 4 °C. The next day, cells were labeled with Alexa Fluor-568 goat anti-mouse for 1 h at RT. Coverslips were mounted in Mowiol^®^.

For the DRG neurite outgrowth assay, embryonic and adult DRG neurons were fixed with 4% PFA and 4% sucrose, and immunostained with Tuj-1-mAb antibodies, as described above. Neurite outgrowth from DRG neurons was quantified by the average of neurite length and the maximal neurite elongation per cell by NeuroJ plugin of ImageJ (FIJI, NIH, https://imagej.net/ImageJ, NIH; accessed on 15 February 2022) (Formoso 2015).

### 2.7. Image Acquisition and Analysis

DRG explants and dissociated DRG neurons were imaged with 10×, 20×, or 60× objectives on an Olympus FV1000 confocal microscope. We set up the Olympus Fluoview v3.1a software to acquire the images with a 4–10 μs/pix of dwell time. We manually adjusted the laser energy setting (HV, gain, and offset) by using slides stained only with the secondary antibodies to determine the threshold of the background signal, which was applied to each image of the experiment. Images were taken in raster-scan mode satisfying the Nyquist criterion; the pixel size was 2–3 times smaller than the object.

For the DRG neurite outgrowth assay, DRG neurons were imaged with an epifluorescence Nikon TE2000-U microscope coupled to an ORCA-ER CCD camera (Hamamatsu), and images were taken with a 60× objective.

### 2.8. Western Blotting

Whole-tissue lysates were prepared in the presence of PIC. Samples containing an equal amount of protein from embryonic and adult DRG were analyzed under reducing conditions in a 10% SDS-PAGE. After electrophoresis, proteins were wet transferred onto a nitrocellulose membrane (Millipore, Burlington, MA, USA) in a tank blot apparatus (Bio-Rad Laboratories, Hercules, California, USA). Membranes were blocked in a Tris-buffered saline (TBS) solution containing 5% non-fat dried milk for 1 h at RT and incubated with primary antibodies diluted in 1% BSA-PBS for 16 h at 4 °C. The next day, membranes were washed with TBS-T (TBS–0.2% Tween 20) and incubated with HRP-conjugated antibodies for 2 h at RT. Antigen-antibody complexes were detected according to a standard enhanced chemiluminescence blotting protocol using a super signal chemiluminescent substrate (ECL-Pierce, Thermo Fisher Scientific, Waltham, MA, USA) and CL–Exposure films (Thermo Fisher Scientific, Waltham, MA, USA).

### 2.9. Bioinformatic Analysis of GPM6a Expression in DRG

To explore the expression of GPM6a’s mRNA within DRGs, we performed a literature search in NCBI PubMed [34], looking for openly available datasets on bulk and single-cell RNA-seq analysis of whole DRG tissues and/or DRG neuronal cells of mouse, rat, and human origins. Our analysis consisted essentially of data mining for GPM6a gene expression levels within publicly available datasets downloaded directly from NCBI GeoDataSets [35,36]. No additional analysis of those datasets was performed. As shown in Table 1, the results are presented as qualitative information (i.e., presence/absence of GMP6a) according to the source tissues in which they were detected, as follows “Whole DRG”, “Nociceptors DRG neurons”, “DRG cultured neurons”, “Satellite glial cells”, and “Sciatic nerve/Schwann cells”.

### 2.10. Statistical Analysis

Calculations were performed with GraphPad Prism 5.0 (San Diego, CA, USA). Data are expressed as mean + SEM. Calculations of two-way ANOVA followed by Bonferroni post-tests were performed. The results were considered significant when *p* < 0.05.

## 3. Results

### 3.1. Bioinformatic Analysis of GPM6a Expression in DRG

DRG neurons can be molecularly distinct, as evidenced by their mRNA-expression signatures [40,48]. There are different kinds of DRG neurons, which are classified according to the type of stimuli they can sense and the molecules they express: nonpeptidergic nociceptors (NP), peptidergic nociceptors (PEP), mechanoreceptors, and proprioceptors [12,40,41]. Therefore, we first performed a bioinformatic analysis of the literature using public bulk or single-cell RNA-seq datasets to analyze the expression of GPM6a’s mRNA within DRGs tissues and cells from mouse, rat, and humans. This information is summarized in Table 1. Our search revealed that GPM6a was expressed in mouse NP-DRG neurons from naïve animals [40] or after a sciatic nerve transection [41]. In the rat, GPM6a expression was detected within the sciatic nerve, the lumbar DRG neurons, and the satellite glia cells (SGC) [39,46]. GPM6a expression was also detected in SGC from mice and humans [46]. Interestingly, GPM6a was detected in rat SCs from the sciatic nerve before and after nerve crushing; however, its expression was a thousand times lower than that of Schwann cell-specific genes [47]. Altogether, we found that the transcript for GPM6a is well expressed in PNS tissues and cells across a range of species, developmental stages, and cell types.

### 3.2. Analysis of the GPM6a Staining Pattern in Rat DRG

#### 3.2.1. IHC of Embryonic and Adult DRGs

As presented here, we aimed to characterize the expression pattern of GPM6a in rat DRGs (undissociated, uncultured). Figure 1 shows representative IHCs of both embryonic (a-b) and adult (c-d) DRGs presenting the first evidence of GPM6a detection in the mammalian PNS. In these images, endogenous GPM6a is shown in magenta; β-tubulin III (Tuj-1-mAb), a neuronal marker, in green; and methyl green staining identifies nuclei in cyan. In embryonic DRGs, GPM6a was detected in the somas of a fraction of Tuj-1 positive neurons (Figure 1a, b and Appendix A). In contrast, in adult DRGs, GPM6a seems to be exhibited by neuronal cell bodies, axons, and possibly also SGCs surrounding Tuj-1-positive neurons (Figure 1c, d and Appendix A). Importantly, GPM6a was immunodetected as a unique band of 35 kDa by Western blot in homogenate samples from embryonic and adult rat DRGs (Appendix A).

#### 3.2.2. Cultures of DRG Explants

We further characterized the detection pattern of GPM6a in an in vitro model of embryonic DRG explants. To that end, rat DRG explants were cultured over a total period of 21 DIV and visualized by immunofluorescence microscopy at 10 and 21 DIV after labeling the explants with GPM6a antibodies together with different cellular markers (Figure 2). S100β antibodies (Figure 2a, in green) were used to label SCs. Antibodies against neurofilament (NFh-mAb, Figure 2b, in green) and Thy-1 (Thy-1-mAb, Figure 2c in green) were both used to label axons. Endogenous GPM6a positive staining (magenta) was observed along the cell bodies (Figure 2c) and the NFh positive and Thy-1 positive axons (Figure 2b,c). On the contrary, we did not observe GPM6a expression in S100β positive SCs (Figure 2a, merge image). In summary, these data suggest that endogenous GPM6a protein was mostly restricted to the neuronal bodies and axons in embryonic DRG explants.

#### 3.2.3. Cultures of Dissociated DRG Neurons

To determine the subcellular distribution of endogenous GPM6a and its role in influencing PNS development, we performed immunofluorescence microscopy analysis of dissociated embryonic DRG neurons cultured between 1 and 5 DIV. Figure 3a–d shows representative images of neurons at different stages of development immunostained for endogenous GPM6a (magenta), Tuj-1 (green), and actin (red). As in the case of the primary cultures of hippocampal neurons [25], GPM6a displayed a non-polarized punctate pattern of expression all over the neuronal cell membrane. GPM6a was detected in the membrane of the soma (Figure 3b–d insets ii) and along the neuronal processes (Figure 3a, insets i and ii, and b–d, inset i). Interestingly, we did not observe positive GPM6a staining in non-neuronal (Tuj-1 negative) cells (F-actin positive cell, Figure 3e). GPM6a staining seemed to be exclusive to Tuj-1 positive DRG neurons, aligned with our IHC analysis of embryonic DRGs. In addition, bulk RNA-seq analysis of rat Schwann cells in culture, treated or not with cAMP (to induce SC differentiation), showed almost no expression of GPM6a when compared with SCs-specific genes (Appendix A, [49]). Altogether, these results indicate that GPM6a is restricted to DRG primary neurons.

After a neurotraumatic injury, a regenerative program is switched on by expressing proteins involved in neuronal development [13]. To reproduce an axonal injury condition, adult DRGs can be harvested in vivo, and then neurons may be dissociated and replaced in a dish (in vitro) [50,51,52]. Thus, immunostainings of adult DRG dissociated neurons in culture were performed to identify whether or not GPM6a might be involved in the post-injury reprogramming. Figure 3f and Appendix A show representative confocal images of adult DRG neurons. As in the case of embryonic neurons, GPM6a staining showed a punctate pattern all over the cell membrane of the neuronal soma and processes (shown in magenta). These results suggest that GPM6a might be involved in PNS repair.

### 3.3. GPM6a Is Involved in Neurite Outgrowth of Cultured DRG Neurons

Monoclonal antibodies against extracellular domains or the N-terminal region of GPM6a have been used to interfere with neurite extension, axon outgrowth, and synapse formation in hippocampal and cerebellar neurons [18,25,27]. Thus, to study if GPM6a contributes to neurite outgrowth in peripheral sensory neurons, embryonic and adult DRG dissociated neurons were treated with neutralizing monoclonal antibodies against GPM6a extracellular domains. Figure 4 shows representative images of control neurons treated with PBS (Figure 4a,f), or experimental neurons treated with 1 µg/mL of GPM6a-mAb (Figure 4b,g) or 3 µg/mL of GPM6a-mAb (Figure 4c, h) for 24 h. Figure 4d–e,i,j show the quantitative image analysis of the average neurite length and the longest distance reached by sensory neurites, showing a significant decrease in neurons treated with GPM6a-mAb. In agreement with what was documented for the CNS neurons, here we demonstrate that GPM6a participates in peripheral sensory neuron development and axon regeneration in vitro.

## 4. Discussion

GPM6a’s expression in CNS neurons has been widely documented to have high abundance in the cortex, hippocampus, and cerebellum [27,53]. Here, we provide the first evidence of GPM6a’s detection in rat DRG neurons by using different experimental approaches, including confocal imaging from immunostained tissues and cultured cells, Western blot analysis, and RNA-seq. In embryonic samples, the expression of GPM6a was confirmed by immunostaining experiments of DRG explants and cultured dissociated DRG cells, and these results correlate with what we found in the RNA-seq bioinformatic analysis [38,39,42]. We found that GPM6a immunostaining was restricted to the plasma membrane of Tuj-1 positive neuronal bodies and in their NFh- and Thy-1- positive axonal projections (Figure 1, Figure 2, Figure 3, and Appendix A). However, specific GPM6a immunostaining was not detected in glial cells from dissociated, cultured DRGs explants or cells (Figure 2 and Figure 3). This was supported by RNA-seq data analysis showing almost no expression of the GPM6a transcript in cultured, isolated rat SCs (Appendix A), and proteomic analysis of mice peripheral myelin showing the lack of GPM6a protein [54,55]. Altogether, these results indicate that, as in the case of CNS neurons, GPM6a is expressed exclusively in PNS neurons and not in glial cells at early stages of development.

We described the presence of GPM6a in the plasma membrane of adult rat DRG neurons in vitro (Figure 1f and Appendix A). This is aligned with reports documenting the expression of the GPM6a-mRNA in DRG neurons from adult mice [38,40,41,44,45]. Satellite glial cells, which form the envelope wrapping each sensory neuron in adult animals, are identified based on their morphologies and locations [56]. Interestingly, our immunohistochemistry studies of whole DRG suggest that GPM6a is also detected in the SGCs (Figure 1c,d, and Appendix A). Moreover, Avraham and colleagues reported GPM6a-mRNA expression in SGCs from lumbar DRGs of adult rats and humans [46].

DRG neurons can be classified depending on the molecules and receptors being expressed. For example, nociceptors are small-caliber unmyelinated or thinly myelinated neurons characterized by expressing the neurotrophin NGF and its receptor, TrkA, together with NFh, TRPV1, substance P (Tac1), and P2RX3 [11,40,41,57]. Proprioceptors are large-caliber, myelinated neurons characterized by expressing NT3 and its receptor TrKC [57]. In particular, Usoskin and colleagues and Hu and colleagues reported the expression of the GPM6a gene in non-peptidergic nociceptors in adult mice [40,41]. In addition, proteomics studies detected GPM6a in the membranes of DRG neurons in a mouse model of chronic and neuropathic pain [58]. Similarly, in the CNS, Cooper and colleagues reported the expression of GPM6a in non-myelinated gluatamatergic axons in adult rat cerebellum [59]. Altogether, these results indicate that GPM6a could be expressed in unmyelinated nociceptive sensory neurons. However, additional studies are needed to identify the presence of GPM6a in specific neuronal subtypes.

As mentioned, an injury to the peripheral nerve triggers a sequence of events that cause the up- and down-regulation of specific proteins [13]. In the CNS, GPM6a was found to be an essential element for neurite outgrowth in several in vitro models [20,21,22,26,27,60] and for axonal elongation in primary neuron cultures using naïve [61] and GPM6a-knockout mice [24]. Here, by utilizing a monoclonal neutralizing antibody against the extracellular domains of GPM6a, which induces GPM6a internalization and degradation [18], we confirmed that GPM6a is required for neurite extension in both embryonic and adult sensory neurons (Figure 4). These results support the idea that GPM6a has a role during PNS development and/or repair of the PNS.

In conclusion, this work shows that GPM6a is present in embryonic and adult rat DRG neurons, and that it participates in neurite extension. Nowadays, a vast amount of data are available from “omic” datasets, and together with the basic research presented here, could help to explain the mechanisms by which the PNS, and particularly, DRG neurons, develop and are able to self-repair. Moreover, our data lay the foundation for follow-up research to study GPM6a in the human PNS in view of designing new therapeutic approaches for neuroprotection or neuroregeneration.

## Figures and Tables

**Figure 1 biomolecules-13-00594-f001:**
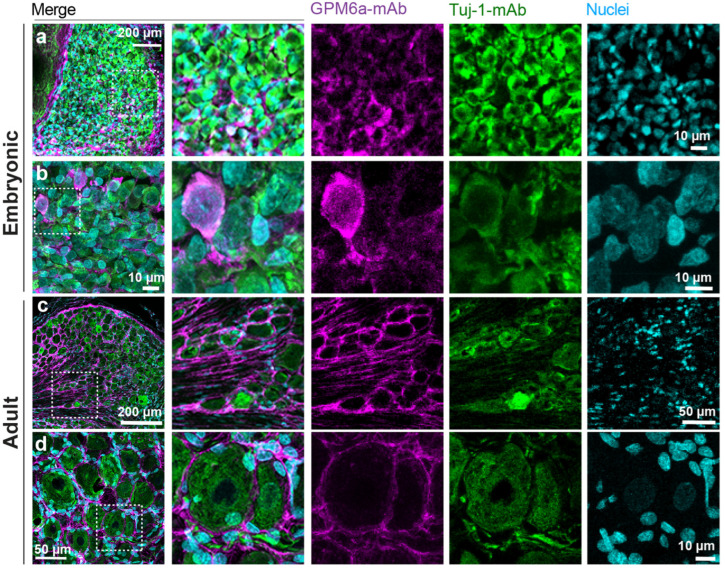
GPM6a was detected in both embryonic and adult rat DRGs. Representative images of embryonic (**a**,**b**) and adult (**c**,**d**) DRGs. N = 3 independent experiments with n = 5–8 DRGs per stage of development were analyzed. Intact DRGs were fixed, prepared for sectioning on a cryostat (at 20 μm), and analyzed immunohistochemically. Images show positive immunostaining for GPM6a (GPM6a-mAb, shown in magenta) and β-tubulin III (Tuj-1-mAb, shown in green). Cell nuclei were imaged with methyl green (cyan). Confocal images were acquired with an Olympus FV1000 confocal microscope using the following settings: (**a**) Ten-times objective and 5× digital zoom. Inset: 80 × 80 µm. (**b**) Sixty-times objective and 2× digital zoom. Inset: 25 × 25 µm. (**c**) Ten-times objective and 2× digital zoom. Inset: 200 × 200 µm. (**d**) Sixty-times objective and no digital zoom. Inset: 65 × 65 µm. Scale bars are indicated in the figure. Images are a maximal projection of 26 z-stacks and 0.5 µm step size (**a**), 22 z-stacks and 0.25 µm step size (**b**), 5 z-stacks and 0.25 µm step size (**c**), and 19 z-stacks and 0.5 µm step size (**d**).

**Figure 2 biomolecules-13-00594-f002:**
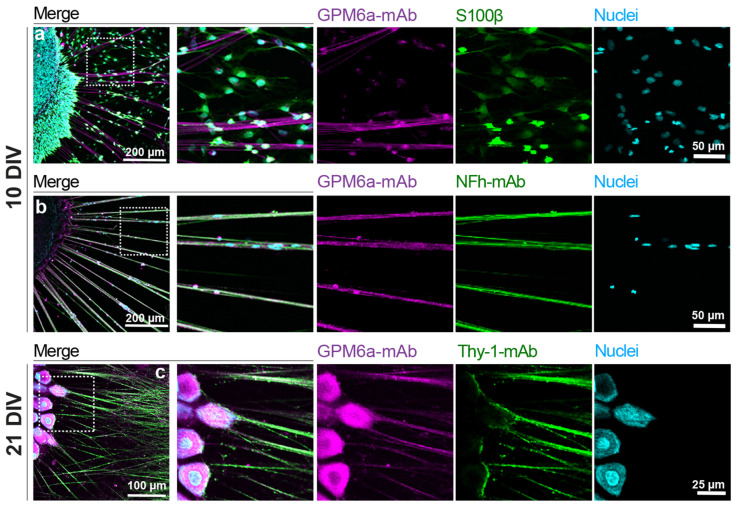
GPM6a’s detection in rat DRG explants. Representative images of embryonic DRGs at 10 DIV (**a**,**b**) and 21 DIV (**c**). N = 2–3 independent experiments with n = 3–8 DRG explants per condition were analyzed. Images show positive immunostaining for GPM6a (GPM6a-mAb, shown in magenta), axons (neurofilament, NFh-mAb, or Thy-1-mAb, shown in green), and SCs (S100β in green). Cell nuclei are shown in cyan. Confocal images were acquired with an Olympus FV1000 confocal microscope using the following settings: (**a**,**b**) 20× objective, insets: 220 × 220 µm; (**c**) 20× objective, inset: 120 × 120 µm. Images from panels (**a**,**b**) are maximal projections of 11 z-stacks and 10 z-stacks respectively, using a step size of 1 µm in both cases. Scale bars are indicated in the figure.

**Figure 3 biomolecules-13-00594-f003:**
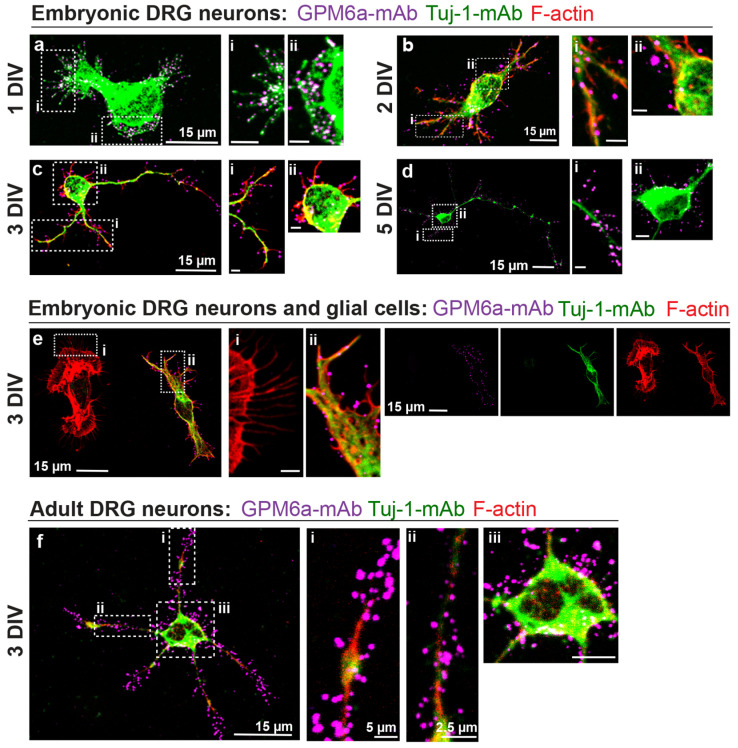
Endogenous GPM6a is expressed in a punctate pattern in the somas and processes of primary cultures of DRG neurons. Representative images of embryonic (**a**–**d**) and adult (**f**) dissociated cultured DRG neurons at different stages of development are shown. N = 3 independent experiments and n = 10–20 cells per condition (DIV, days in vitro) were analyzed. Each experiment had at least three wells per conditions (n = 3 technical replicates). Images show positive immunostaining for GPM6a in magenta (GPM6a-mAb), β-tubulin III in green (Tuj-1-mAb), and actin in red (F-actin). GPM6a expression was within the neuronal processes (insets i for embryonic and insets i/ii for adult neurons) and the soma (insets ii for embryonic and inset iii for adult neurons). Confocal images were acquired using a 60× objective and 4× (**a**), 2× (**b**), 3× (**c**, **f**), and 1.3× (**d**) digital zoom. Images are maximal projections of 4 z-stacks (**a**), 3 z-stacks (**b**), 3 z-stacks (**c**), 6 z-stacks (**d**), and 4 z-stacks (**f**) and 0.15 µm step size (**a**–**d**, **f**). Scale bars for the insets represent 5 µm. (**e**) Representative confocal images of embryonic dissociated DRG neurons depicting contaminating glial cells at 3 DIV. Cells were immunostained with Tuj-1-mAb (green) and F-actin (red) to differentiate neurons from other cells. Endogenous GPM6a was detected with GPM6a-mAb antibodies (magenta). GPM6a expression was observed only within neurons (ii) and not glial cells (i). Insets: 20 × 10 µm. Confocal images were acquired using a 60× objective and 2× digital zoom. Images are maximal projections of 4 z-stacks and 0.15 µm step size. All confocal images were acquired with an Olympus FV1000 confocal microscope.

**Figure 4 biomolecules-13-00594-f004:**
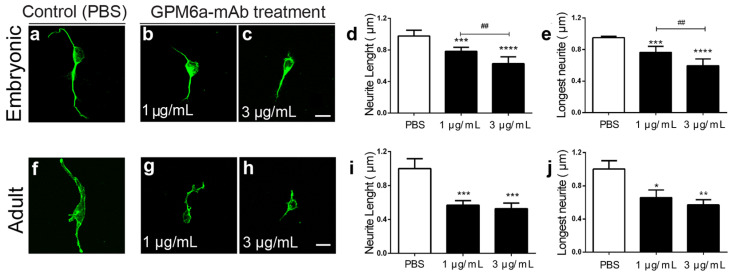
Blocking endogenous GPM6a impairs neurite elongation in cultured DRG neurons. Representative images show embryonic and adult dissociated DRG neurons treated (**b**,**c**,**g**,**h**) or not (**a**,**f**, control condition) with the neutralizing GPM6a-mAb antibody for 24 h. DRG neurons were stained with Tuj-1-mAb (green). Scale bars represent 10 µm. The neurite length (**d**,**i**) and the longest neurite (**e**,**j**) were quantified with ImageJ software using the NeuronJ plugin. Results are expressed as mean + SEM of N = 2–4 independent experiments with 15–25 cells per condition. Each experiment had at least three wells per condition (n = 3 technical replicates). Two-way ANOVA followed by Bonferroni post-test calculations were used. The results were considered significant when *p* < 0.05. * *p*< 0.05, ** *p* < 0.01, *** *p* < 0.005, and **** *p* < 0.001 for PBS vs. GPM6a-mAb, 1 μg/mL, or 3 μg/mL; and ## *p* < 0.01 for GPM6a-mAb 1 μg/mL vs. GPM6a-mAb 3 μg/mL.

**Table 1 biomolecules-13-00594-t001:** Expression of GPM6a’s mRNA in the DRG and other PNS-derived tissues and cells from mice, rats and humans. Bioinformatic analysis of GPM6a’s gene expression from bulk or single-cell RNA-seq from public datasets. All DRGs samples or cultured neurons were localized to the lumbar area with the exception of those reported by Sharma et al. 2020, which did not specify the anatomical locatiosn of the samples. ***** Sharma et al. 2020 analyzed DRGs at different stages of development: E11.5, E12.5, E15.5, P0, P5 and P28-42. **+**: Expression of GPM6a gene in the indicated samples. Downward arrow means downregulation of the GPM6a transcript. DRG: dorsal root ganglia, SCs: Schwann cells, E: embryonic day, and P: postnatal day.

Result	Species	Reference
**+** Whole DRGs.	Human	Flegel et al., 2015 [37]
	Adult Mice	Wangzhou et al., 2020 [38]
	Adult Rats	Sapio et al., 2020 [39]
**+** Nociceptor DRG neurons.	Adult Mice	Usoskin et al., 2014 [40]
		Hu et al., 2016 [41]
	Embryonic, young and adult Mice *****	Sharma et al., 2020 [42]
**+** DRG cultured neurons.	Adult Mice	Lerch et al., 2012 [43]
		Li et al., 2016 [44]
		Lopes et al., 2017 [45]
		Wangzhou et al., 2020 [38]
	Human	Wangzhou et al., 2020 [38]
**+** Satellite glial cells.	Human Adult Mice	Avraham et al., 2022 [46]
	Adult Rats	
**+** Sciatic nerve/SCs. **↓** upon nerve crush	Rats	Brosius-Lutz et al., 2022 [47]

## Data Availability

Not applicable.

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
