# Peer review of "Endogenous Glycoprotein GPM6a Is Involved in Neurite Outgrowth in Rat Dorsal Root Ganglion Neurons"

_biomolecules, 2023, doi:10.3390/biom13040594_

Round 1
Reviewer 1 Report
This study describes for the first time the localization of the glycoprotein M6a in the DRG during development and in the adult. M6a is mainly neuronal in the embryonic DRG whereas in the adult DRG it is mainly/only found in satellite cells (although this is not entirely clear) but upregulated in adult DRG neurons in culture during neurite outgrowth. Although confocal microscopy was used, the exact subcellular localization of M6a in the neuronal soma is not clear and should be clarified. Experiments about the effects of M6a inhibition on neurite outgrowth have also to be performed in adult DRG cultures.
Lines 36-40: “The peripheral nervous system (PNS) has the unique capacity for functional recovery and self-repair after an injury, a feature seemingly missing from the central nervous system (CNS) [1, 2]. Upon nerve injury, two main programs take place: the ability of the PNS glial cells to promote neuroprotection, regeneration, and re-myelination [3, 4] and the capacity of PNS neurons to re-initiate axon outgrowth [1].”
Reference 1 is not correct in this context: Quarta, S., et al., Peripheral nerve regeneration and NGF-dependent neurite outgrowth of adult sensory neurons converge on STAT3 phosphorylation downstream of neuropoietic cytokine receptor gp130. J Neurosci, 2014. 34(39): p. 13222-33.
Lines 40-43: “The regenerative properties of the PNS have been used clinically to develop therapies to promote neuroprotection and regeneration in the CNS, such as to treat trauma after spinal cord injury and to arrest neurodegeneration in Parkinson´s or Alzheimer´s disease [5-8].”
“have been used clinically” implies that it is used as a clinical therapy.
Lines 94-94: “Dataset libraries such as BioGPS1 and the Human Protein Atlas2 show the expression of the human GPM6A gene in DRG neurons.”
Where are the references for BioGPS1 and the Human Protein Atlas2?
Lines 226-228: “First, we analyzed the expression of M6a in embryonic and adult DRGs. Figure 1 shows representative IHCs of both embryonic (a-b) and adult (c-d) rat DRGs, presenting the first evidence of M6a expression in the PNS.”
The term expression stands for gene expression and is not correct for IHC which shows the protein.
A short description how the bioinformatics analysis was performed is required.
Lines 230-231: “In embryonic DRGs, M6a is distributed in the plasma membrane of a fraction of Tuj-1 positive neurons (Fig 1a-b and Video S1).”
How do the authors know that M6a is exactly distributed in the plasma membrane? In Fig. 2 it looks more like a cytoplasmic staining. How is the extact subcellular localization of M6s in the soma of DRG neurons? This has to be clarified under the different conditions.
It is not clear how many experiments were performed in Fig. 1-3 and how many cells were analyzed in each experiment.
How was the adult, dissociated DRG culture performed? Was it the same for embryonic and adult cultures because adult rat DRG have stronger connective tissue. Was collagenase or liberase used?
The regenerative capacity of embryonic and adult DRG neurons differs substantially. Thus, the effects of M6a inhibition on axon growth of adult DRG neurons have to be evaluated.
"Figure S1: M6a expression in DRGs is restricted to neurons.”
Whole-tissue DRG lysates contain not only neurons but also glia and connective tissue. This conclusion is not correct. Furthermore, it contradicts the IHC of adult DRG.
Author Response
1-This study describes for the first time the localization of the glycoprotein M6a in the DRG during development and in the adult. M6a is mainly neuronal in the embryonic DRG whereas in the adult DRG it is mainly/only found in satellite cells (although this is not entirely clear) but upregulated in adult DRG neurons in culture during neurite outgrowth. Although confocal microscopy was used, the exact subcellular localization of M6a in the neuronal soma is not clear and should be clarified. Experiments about the effects of M6a inhibition on neurite outgrowth have also to be performed in adult DRG cultures.
We understand your concerns regarding GPM6a´s detection in adult DRG neurons and its localization. To clarify these issues, we added a new figure (S2) showing two adult DRG neurons at 2 and 3 DIV with GPM6a positive immunostaining at the cell membrane. Our immunofluorescence workflow for dissociated neurons involves different steps in which we start to label GPM6a with GPM6a-mAb antibodies, which recognizes the extracellular loops of the protein, in living neurons at 4 ºC. Then we proceed with the fixation, permeabilization and blocking steps followed by Tuj-1 and actin staining (see Immunofluorescence in dissociated DRG neurons in material and methods section, page 4, lines 172-178). In other words, under our labeling conditions GPM6a staining is only restricted to the cell membrane.
We also incorporated GPM6a-mAb inhibition on neurite outgrowth in adult DRG cultures in Figure 4. Indeed, we found a significantly decrease in neurite elongation in those neurons treated with 1 or 3 µg/ml of GPM6a-mAb compare to control group.
We agree with the reviewer in that the confocal imaging of whole DRG neurons does not allow us to discriminate glial versus neuronal GPM6a immunostaining. As shown in Fig 2S (right panel, white arrow), we have not found GPM6a immunoreactivity in non-neuronal, Tuj-1 negative cells in cultured adult DRG dissociated cells. This result was confirmed in cultures from embryonic DRGs (Fig 3e) and cultures from purified, isolated Schwann cells (FigS1,b). However, having in mind those results and the IHC of adult DRG we do not exclude the presence of GPM6a in satellite glial cells. Thus, the statement at page 6, lines 265-269 was modifies as follows:
“By contrast, in adult DRGs GPM6a seems to be exhibited by neuronal cell bodies, axons, and possibly also SGCs surrounding Tuj-1 positive neurons (Figure 1c-d and Video S2)”
2-Lines 36-40: “The peripheral nervous system (PNS) has the unique capacity for functional recovery and self-repair after an injury, a feature seemingly missing from the central nervous system (CNS) [1, 2]. Upon nerve injury, two main programs take place: the ability of the PNS glial cells to promote neuroprotection, regeneration, and re-myelination [3, 4] and the capacity of PNS neurons to re-initiate axon outgrowth [1].”
Reference 1 is not correct in this context: Quarta, S., et al., Peripheral nerve regeneration and NGF-dependent neurite outgrowth of adult sensory neurons converge on STAT3 phosphorylation downstream of neuropoietic cytokine receptor gp130. J Neurosci, 2014. 34(39): p. 13222-33.
Thank you, the reference has been removed in that statement in the new version of revised manuscript.
3-Lines 40-43: “The regenerative properties of the PNS have been used clinically to develop therapies to promote neuroprotection and regeneration in the CNS, such as to treat trauma after spinal cord injury and to arrest neurodegeneration in Parkinson´s or Alzheimer´s disease [5-8].”
“have been used clinically” implies that it is used as a clinical therapy.
We agree with you, we meant to expressed that cell therapies strategies to promote neuroprotection and regeneration have been explored to treat trauma after a spinal cord injury or neurodegenerative diseases like Parkinson´s disease (PD) in Phase I clinical trials (Anderson 2017, van Horne 2017, van Horne 2018). For instance, in the clinical trial NCT01739023, investigators evaluated the safety of autologous human Schwann cells transplantation into the injury epicenter of subjects with subacute spinal cord injury. Another example is the clinical trials conducted by Dr. van Horne and colleagues where they implanted an autologous peripheral nerve graft into the substantia nigra of PD patients undergoing deep brain stimulation surgery (NCT0183364 and NCT01739023). In these cases, the safety of the clinical trials was evaluated and no surgical, medical or neurological complications were observed after 1 year follow up. However, we understand the confusion of the sentence and now the word clinically was replaced by explored (please see below) and the references to these clinical trials have been included in the manuscript (page 2, lines 40-42).
“The regenerative properties of PNS cells have been exploited to 40 develop therapies to treat CNS trauma after spinal cord injury and prevent neurodegeneration in Parkinson´s disease [5-11].”
Added References:
Anderson KD, Guest JD, Dietrich WD, et al. Safety of Autologous Human Schwann Cell Transplantation in Subacute Thoracic Spinal Cord Injury. J Neurotrauma. 2017;34(21):2950-2963. doi:10.1089/neu.2016.4895
C.G. van Horne, J.E. Quintero, J.A. Gurwell, R.P. Wagner, J.T. Slevin, and G.A. Gerhardt, Implantation of autologous peripheral nerve grafts into the substantia nigra of subjects with idiopathic Parkinson's disease treated with bilateral STN DBS: a report of safety and feasibility. Journal of neurosurgery 126 (2017) 1140-1147.
C.G. van Horne, J.E. Quintero, J.T. Slevin, A. Anderson-Mooney, J.A. Gurwell, A.S. Welleford, J.R. Lamm, R.P. Wagner, and G.A. Gerhardt, Peripheral nerve grafts implanted into the substantia nigra in patients with Parkinson's disease during deep brain stimulation surgery: 1-year follow-up study of safety, feasibility, and clinical outcome. Journal of neurosurgery 129 (2018) 1550-1561.
4-Lines 94-94: “Dataset libraries such as BioGPS1 and the Human Protein Atlas2 show the expression of the human GPM6A gene in DRG neurons.”
Where are the references for BioGPS1 and the Human Protein Atlas2?
Both references have now added to the revised manuscript as a footnotes the website of each dataset libraries and the reference in the “Reference section” of the manuscript (Page 2, line 84 and page 3, line 85).
5-Lines 226-228: “First, we analyzed the expression of M6a in embryonic and adult DRGs. Figure 1 shows representative IHCs of both embryonic (a-b) and adult (c-d) rat DRGs, presenting the first evidence of M6a expression in the PNS.”
The term expression stands for gene expression and is not correct for IHC which shows the protein.
Base on your comment we use “GPM6a detection” or synonyms instead of “GPM6a expression” throughout the manuscript.
6-A short description how the bioinformatics analysis was performed is required.
A brief description of bioinformatics analysis of the public data sets has been added to material and methods section (page 5, lines 215 to 225):
Bioinformatics analysis of GPM6a expression in DRG 210
To explore the expression of GPM6a´s mRNA within DRGs, we performed a literature search in NCBI PubMed [35] looking for openly available data sets on bulk and single-cell RNA-seq analysis of whole DRG tissues and/or DRG neuronal cells of mouse, rat, and human origins. Our analysis consisted essentially on data mining for GPM6a gene expression levels within publicly available datasets downloaded directly from NCBI GeoDataSets [36, 37]. No additional analysis of those datasets was performed. As shown in Table 1, the results were presented as qualitative information (i.e., presence/absence of GMP6a) according to the source tissue in which it was detected, as follows “Whole DRG”, “Nociceptors DRG neurons”, “DRG cultured neurons”, “Satellite glial cells”, and “Sciatic nerve/Schwann cells”.
References:
35) Edgar R, Domrachev M, Lash AE. Gene Expression Omnibus: NCBI gene expression and hybridization array data repository. Nucleic Acids Res. 2002;30(1):207-210. doi:10.1093/nar/30.1.207
36) Barrett T, Wilhite SE, Ledoux P, et al. NCBI GEO: archive for functional genomics data sets--update. Nucleic Acids Res. 2013;41(Database issue):D991-D995. doi:10.1093/nar/gks1193
37) Sayers EW, Bolton EE, Brister JR, et al. Database resources of the national center for biotechnology information. Nucleic Acids Res. 2022;50(D1):D20-D26. doi:10.1093/nar/gkab1112
7-Lines 230-231: “In embryonic DRGs, M6a is distributed in the plasma membrane of a fraction of Tuj-1 positive neurons (Fig 1a-b and Video S1).”
How do the authors know that M6a is exactly distributed in the plasma membrane? In Fig. 2 it looks more like a cytoplasmic staining. How is the exact subcellular localization of M6s in the soma of DRG neurons? This has to be clarified under the different conditions.
We agree with you, based on our IHC workflow we cannot distinguish the subcellular localization of GPM6a. The statement “M6a is distributed in the plasma membrane fraction of tuj-1 positive neurons” was removed in the revised manuscript version” (page 6 line 264). Also, to label GPM6a in DRG explants we included a permeabilization step for better penetration of the antibodies into the tissue. Therefore, the “cytoplasmic staining of GPM6a” in Figs. 1 and 2 can be due to: a) low quality of figure in the pdf file for reviewers and/or b) excessive tuning of brightness and contrast levels in the images processing with Image J software and/or c) the presence of GPM6a at the membrane of endosomal compartments that we have already documented in Garcia et al. 2017. We now added a statement in page 4, lines 171-178 explaining in more detailed the protocol for immunostaining of DRG explants in culture:
“In the case of embryonic DRG explants, they were fixed in 4% PFA-PBS at 10 or 21 DIV (days in vitro), (Figure 2) before permeabilizing and blocking as described above. Subsequently, DRG explants were incubated with GPM6a-mAb and NFh-mAb or Thy-1-mAb or S100β antibodies in 3% BSA for 16 h at 4 ºC. After, explants were rinsed with PBS and labelled with the corresponding pre-absorbed secondary antibodies for 1 h at RT. Cell nuclei were stained with Dapi and were mounted with coverslips in Mowiol®.”
We also want to clarify that in the IHC and immunofluorescences we set up the confocal microscope software to acquire the images as follows (page 5, lines 192 to 199):
- 4–10 μs/pix of dwell time
-We manually adjusted the laser energy setting (HV, gain, and offset) by using slides stained only with the secondary antibodies to determine the threshold of the background signal, which was applied to each image of the experiment.
- Images were taken in raster scan mode satisfying the Nyquist criterion, pixel size was 2–3 times smaller than the object.
In other words, under these conditions we assure that the GPM6a immunostaining is specific.
Reference:
Garcia MD, Formoso K, Aparicio GI, Frasch ACC, Scorticati C. The Membrane Glycoprotein M6a Endocytic/Recycling Pathway Involves Clathrin-Mediated Endocytosis and Affects Neuronal Synapses. Front Mol Neurosci. 2017;10:296. Published 2017 Sep 20. doi:10.3389/fnmol.2017.00296
7-It is not clear how many experiments were performed in Fig. 1-3 and how many cells were analyzed in each experiment.
The number of experiment and cells analyzed were now added to the revised manuscript at the legend of each figure.
8-How was the adult, dissociated DRG culture performed? Was it the same for embryonic and adult cultures because adult rat DRG have stronger connective tissue. Was collagenase or liberase used?
You are right the dissociated adults DRG neurons differ with the embryonic DRG neurons. Thus we added the complete protocol in the material and methods section (page 4, lines 135 to 155).
Isolation and culture of dissociated DRG neurons from adult rats
DRGs and dissociated DRG neurons from 6-8 weeks old Sprague Dawley rats were used for Figure 1c-d, Figure 3f and Supplemental Figure S2a-b. Both male and female animals were used in an approximately equal number for all of the experiments. Each experiment contained a minimum of three independent experiments. Adult DRG neurons culture was prepared according to the method described by Burkey et al. [31]. Briefly, rats were sacrificed and the DRGs were dissected and collected in ice-cold modifies Hank´s balance salt solution (HBSS, 171 mM NaCl, 6.7 mM KCL, 1.6 mM Na2PO4, 05.5 mM KH2PO4, 6mM D-glucose, and pH 7.3) supplemented with 50 U/mL Penicillin/Streptomycin (Sigma). HBSS were carefully aspirated and replaced with a pre-warm collagenase type 1A solution (1 ml per 6 isolated DRGs) and incubated 30 min at 37 ºC. Then, we carry out an additional digestion for 30 min in 1 ml of 0.25% trypsin in HBSS at 37 ºC according to Kim et al. [32].The supernatant was aspirated and DRG pellet were resuspended in adult growth medium (AGM, Ham´s F12 medium (GIBCO) with 10 % of horse serum supplemented with 2mM L-glutamine and 50U/mL Penicillin/Streptomycin and dissociated by gentle mechanical agitation using a glass Pasteur pipet until the suspension is homogenous. Cells were pellet by centrifugation at 200 g for 10 min and added AGM with NGF (250 ng/mL) for cell counting. Cells were seeded on coverslips coated with PLL-Laminin at a low density of 10-20x104 cells/well in a 24-well plate. Cells were maintained at 37 ºC and 5%CO2. Medium were replaced every 2 days.
References
1-Burkey TH, Hingtgen CM, Vasko MR. Isolation and culture of sensory neurons from the dorsal-root ganglia of embryonic or adult rats. Methods in Molecular Medicine. 2004 ;99:189-202. DOI: 10.1385/1-59259-770-x:189. PMID: 15131338.
2- Kim HW, Davies AJ, Oh SB. In Vitro Visualization of Cell-to-Cell Interactions Between Natural Killer Cells and Sensory Neurons. Methods Mol Biol. 2022;2463:251-268. doi: 10.1007/978-1-0716-2160-8_18. PMID: 35344180.
9-The regenerative capacity of embryonic and adult DRG neurons differs substantially. Thus, the effects of M6a inhibition on axon growth of adult DRG neurons have to be evaluated.
As mentioned above, we incorporated new data showing the GPM6a-mAb inhibition on neurite outgrowth in adult DRG cultures (Figure 4). Indeed, we found a significantly decrease in neurite elongation in those neurons treated with 1 or 3 µg/ml of GPM6a-mAb compare to control group.
10-"Figure S1: M6a expression in DRGs is restricted to neurons.”
Whole-tissue DRG lysates contain not only neurons but also glia and connective tissue. This with is not correct. Furthermore, it contradicts the IHC of adult DRG.
We totally agree we now replaced Figure S1 tittle as follows “GPM6a is detected in DRGs homogenates but not in Schwann Cells”.
Reviewer 2 Report
The authors characterized the expression of neuronal membrane glycoprotein M6-a (GPM6A) in embryonic and adult dorsal root ganglion (DRG) by combining RNA-seq public data sets with immunochemical approaches utilizing cultures of rat DGR explants and neuronal cells. It’s very interesting that GMM6A is expressed in DRGs and at the cell surface of cultured DRG neurons. Additionally, GPM6A is expressed at the membrane of satellite glia cells in the DRGs of adults. Moreover, the authors found that GPM6A is required for neurite elongation in DRG neurons. Although this is a primary research for GMM6A in morphology, it’s very important for the authors or others to further study.
However, there are several minor points need to be addressed before publication.
1. In order to be searched and identified M6a, I suggest the authors use GPM6A as the abbreviation instead of M6a. Because N6-methyladenosine is also abbreviated as m6A.
2. In Figure 3f, the last image should be iii instead of ii on the left corner.
3. In line 236, there is no need a coma after “both”.
4. In the figure legends of Figure 4, suggest the authors point out that *** is compared between PBS and 1μg/ml or 3μg/ml (***: PBS VS. 1μg/ml or 3μg/ml). ## is compared between 1μg/ml and 3μg/ml (##:1μg/ml VS. 3μg/ml).
Author Response
- In order to be searched and identified M6a, I suggest the authors use GPM6A as the abbreviation instead of M6a. Because N6-methyladenosine is also abbreviated as m6A.
During these years we have followed Uniprot/NCBI's rules for taxonomy and nomenclature of genes and proteins (e.g https://www.uniprot.org/uniprotkb/P51674/entry), however, we agree and understand your and the editor´s concerns about GPM6a and N6-methyladenosine names and abbreviations. Thus, we replaced M6a with GPM6a in the entire manuscript.
- In Figure 3f, the last image should be iii instead of ii on the left corner.
Thank you for your suggestion, the Figure 3f has been modified accordingly.
- In line 236, there is no need a coma after “both”.
Thank you, the coma in the figure legend 1 was removed.
- In the figure legends of Figure 4, suggest the authors point out that *** is compared between PBS and 1μg/ml or 3μg/ml (***: PBS VS. 1μg/ml or 3μg/ml). ## is compared between 1μg/ml and 3μg/ml (##:1μg/ml VS. 3μg/ml).
Thank you for your suggestion, the statistical comparison in the legend Figure 4 has been added.
Round 2
Reviewer 1 Report
The manuscript improved substantially in the revised version.
Minor comments:
2-Lines 36-40: “The peripheral nervous system (PNS) has the unique capacity for functional recovery and self-repair after an injury, a feature seemingly missing from the central nervous system (CNS) [1, 2]. Upon nerve injury, two main programs take place: the ability of the PNS glial cells to promote neuroprotection, regeneration, and re-myelination [3, 4] and the capacity of PNS neurons to re-initiate axon outgrowth [1].”
Reference 1 is not correct in this context: Quarta, S., et al., Peripheral nerve regeneration and NGF-dependent neurite outgrowth of adult sensory neurons converge on STAT3 phosphorylation downstream of neuropoietic cytokine receptor gp130. J Neurosci, 2014. 34(39): p. 13222-33.
Thank you, the reference has been removed in that statement in the new version of revised manuscript.
The reference was not removed but transferred to ref. 4 but it is still not correct in the context. Could the authors please explain why they used this reference again.
4-Lines 94-94: “Dataset libraries such as BioGPS1 and the Human Protein Atlas2 show the expression of the human GPM6A gene in DRG neurons.”
Where are the references for BioGPS1 and the Human Protein Atlas2?
Both references have now added to the revised manuscript as a footnotes the website of each dataset libraries and the reference in the “Reference section” of the manuscript (Page 2, line 84 and page 3, line 85).
Where are the footnotes of the websites?
Author Response
1-The reference was not removed but transferred to ref. 4 but it is still not correct in the context. Could the authors please explain why they used this reference again.
We have re-checked the reference the reviewer is referring to (Quarta, S., et al. 2014) and we agree that it is not correct in this context. The paragraph in question, (2-Lines 36-40) is referring to the capacity of the PNS of recovery and self-repair after an injury in a broad manner. In this sense, Quarta, S., et al. 2014 does not focus on this subject but rather in the role of NGF-pathway on neurite outgrowth in sensory neurons. Therefore, we now removed completely this reference from the re-revised version of the manuscript.
4-Lines 94-94: “Dataset libraries such as BioGPS1 and the Human Protein Atlas2 show the expression of the human GPM6A gene in DRG neurons.”
Where are the footnotes of the websites?
We believe there must be a misunderstanding, both references of the first description of each data set library have been already added at the references list in the revised version of the manuscript. However, in order to prevent PDF edition errors that could occur, we added the websites links in the main text instead of in the footnotes as follows (page 2, lines 84-85):
Dataset libraries such as BioGPS [http://biogps.org, 30] and the Human Protein Atlas [https://www.proteinatlas.org, 31] show...
